# Clinical experiences, current approaches, opinions and awareness of healthcare professionals regarding the audio-vestibular consequences of individuals with traumatic brain injury: a cross-sectional online survey study

Kübra Bölükbaş ![ORCID],[1,2] Laura Edwards ![ORCID],[3,4] David M Baguley ![ORCID],[1,2] Kathryn Fackrell ![ORCID][1,2,5]

For numbered affiliations see end of article.

**Correspondence to**
Kübra Bölükbaş;
kubra.bolukbas@nottingham.ac.uk

## ABSTRACT

**Objective** To explore the experiences, current approaches, opinions and awareness of healthcare professionals (HCPs) caring for adults with traumatic brain injury (TBI) regarding the audio-vestibular consequences.

**Design/setting** Cross-sectional online survey study.

**Participants** HCPs with experience of caring for adults with TBI, who were not ENT (ear nose throat) specialists or audiologists.

**Methods** The study was conducted from May 2022 to December 2022. The online survey consisted of 16 closed and open-text questions in English and Turkish about clinical experience, current approaches and awareness of audio-vestibular consequences following TBI. Frequencies of responses to closed questions and associations between variables were analysed using SPSS V.28. Open-text responses were summarised in Microsoft Excel.

**Results** Seventy HCPs participated from 17 professions and 14 countries, with the majority from the UK (42.9%). HCPs stated that 'some' to 'all' of their patients had auditory problems such as 'inability to understand speech-in-noise' (66%), 'tinnitus' (64%), 'hyperacusis' (57%) and balance problems such as 'dizziness' (79%) and 'vertigo' (67%). Usually, HCPs asked about the balance status of patients at appointments and when they observed dizziness and/or balance disorder they used screening tests, most commonly finger-to-nose (53%). For auditory impairments, HCPs preferred referring patients with TBI to audiology/ENT services. However, 6% of HCPs felt that audio-vestibular conditions could be ignored on referral because patients with TBI struggled with many impairments. Additionally, 44% would suggest hearing aids to patients with TBI with hearing loss 'if they would like to use' rather than 'definitely'.

**Conclusions** Many audio-vestibular impairments are observed by HCPs caring for patients with TBI. The assessment and intervention opinions and awareness of HCPs for these impairments vary. However, non-expert HCPs may not be aware of negative consequences of untreated audio-vestibular impairments following TBI. Therefore, developing a simple framework for screening

## STRENGTHS AND LIMITATIONS OF THIS STUDY

⇒ The survey was conducted online which facilitated the participation of healthcare professionals (HCPs) from diverse geographical regions worldwide.

⇒ The multidisciplinary approach to recruitment using established academic networks resulted in a range of respondents from different health disciplines.

⇒ The creation of two versions of the survey enhanced inclusivity and made it possible for respondents especially from Türkiye to participate who may not be proficient in English.

⇒ Despite our efforts to engage HCPs from across the world the majority of respondents were from the UK and as such findings may reflect the UK health system.

and indications of audio-vestibular impairments for referral may be helpful for non-audiological specialists regularly seeing these patients.

## INTRODUCTION

Traumatic brain injury (TBI) is defined as a structural injury and/or physiological deterioration of brain function following trauma due to an external force.[1] It is estimated that 64–74 million people every year experience a TBI[2] and it is one of the leading causes of death and disability worldwide.[3 4] There are many different complications associated with TBI, such as cognitive, emotional, behavioural and audio-vestibular impairments.[5–8] In addition, neck pain, balance disorders and tinnitus (ringing in the ear(s)) have been reported as some of the most common complaints after head and neck trauma.[9 10] Moreover, in a review investigating hearing loss in TBI without skull fracture, Chen *et al*[11] found that

58% of patients with TBI experienced hearing loss, sometimes temporarily, despite not having any skull fractures. This rate suggests that the percentage of audio-vestibular consequences related to TBI that are not reported and/or investigated worldwide may be much higher than expected.

Consequences of TBI vary greatly by nature, severity and duration. Some individuals may be left with minimal impairment and make a rapid recovery with/without specialist intervention, while others may require inpatient or outpatient neurorehabilitation.[1] This diversity of complications due to TBI necessitates a multidisciplinary approach. Accordingly, depending on the severity and nature of the complication, a patient with TBI may be treated by a wide range of healthcare professionals (HCPs) including physiotherapists, psychologists and occupational therapists. The emergence of hearing loss and/or vestibular disorders, tinnitus or hyperacusis (sound sensitivity) following TBI further complicates the treatment of TBI and requires a sensitive and careful approach in the diagnosis and treatment of audio-vestibular disorders in this patient group. Untreated hearing loss for example can lead to additional conditions such as dementia[12] as well as many psychological problems such as communication difficulties and social isolation.[13 14] Therefore, assessment and treatment of audio-vestibular conditions should not be neglected when caring for adults with TBI as early detection of audio-vestibular impairments in patients after TBI can improve quality-of-life, minimise complications and prevent additional conditions such as dementia.

Patients struggling with many problems such as physical, cognitive, emotional and behavioural complications after TBI may not easily notice any audio-vestibular impairments immediately. However, HCPs who care for them should understand and assess their audio-vestibular conditions, even if it is not their field of expertise and, if necessary, arrange referrals to the ENT (ear nose throat) specialists or audiologists. A recent survey on audio-vestibular clinical practice within Australian audiology departments found gaps in audiologist knowledge and inadequate resources and practice regarding the identification, diagnosis and management of patients with auditory and/or vestibular deficits following TBI.[15] However, as yet, the clinical experiences, current approaches, opinions and awareness of HCPs who specialise in managing patients with TBI have not been fully investigated. It is therefore important to understand the experiences of these HCPs in particular to understand their awareness and knowledge of the audio-vestibular consequences of TBIs, in order to understand and identify areas that could be improved in existing practices.

Accordingly, the aim of this study was to explore and gain an understanding of the clinical experiences, current approaches, opinions and awareness of the HCPs caring for adults with TBI (except ENT specialists and audiologists) regarding the audio-vestibular consequences experienced by these patients.

## METHODS
### Design
This study used a cross-sectional international online survey to explore the clinical experiences, current approaches, opinions and awareness of HCPs regarding the audio-vestibular conditions experienced by patients who had a TBI. Research approval was obtained from the University of Nottingham, Faculty of Medicine and Health Sciences Research Ethics Committee (FMHS 462-0222, May 2022). Before initiating the survey questions, on the first page, an information page was provided detailing who the researchers are, the purpose of the survey, its duration, the advantages and disadvantages of participation, who would have access to the survey data and where and for how long the data would be stored. Following the information page, all participants were requested to give informed consent. This survey study is reported according to the Checklist for Reporting Results of Internet E-Surveys (CHERRIES).[16]

### Survey development
The survey questions were iteratively developed and refined through discussion and consensus within the study team. The authors initially compiled a list of potential questions and relevant sections for inclusion in the survey. Questions were based on experience of the authors and evidence from research. To ensure the questions and response options were relevant and valid for HCPs, the surveys were piloted with two HCPs and one researcher who have experience of audio-vestibular assessments. The questions were revised and distilled down to create a 16-question survey (over 10 pages, with a maximum of 3 questions per page) which included closed questions with single-choice, multiple-choice options or Likert scales and for four questions with a free-text 'other' option. Subquestion (16a) of the last question used adaptive questioning, which was only displayed conditionally based on 'yes' responses to question 16. Respondents were asked to consider patients with TBIs that occurred more than 6 months ago and the reported or observed audio-vestibular symptoms at that time. It was necessary to answer each question and to complete the survey in one session. It was possible for respondents to review and edit their responses until the point they chose to submit the survey on the final page. In order to reach a range of HCPs, and since the lead author (KB) is bilingual, two versions of the survey were created: one in English and one in Turkish (online supplemental appendix 1). The survey was delivered online using JISC software maintained by the University of Nottingham (https://www.onlinesurveys.ac.uk/) which collected and stored responses from each respondent. Respondents accessed the open survey via a link (passwords were not provided).

### Patient and public involvement
No patients were directly involved in the development, dissemination or reporting of this study. The study focused on the experiences of HCPs in the field of TBI,

such as physiotherapists, neuropsychologists, rehabilitation nurses, therefore the survey was piloted with HCPs and a researcher who have experience of audio-vestibular assessments to gain their views and refine questions, and two of the authors are clinicians with relevant experience in the field.

## Study sample and eligibility

The study population was identified as HCPs caring for adult patients with TBI worldwide. To be eligible to participate in the survey, respondents were required to be HCPs with clinical experience in caring for adult patients with TBI and to have the ability to understand and answer questions in English (except HCPs in Türkiye). ENT specialists and audiologists were excluded from the study as they were already experts in the field of audio-vestibular care.

## Distribution

A number of recruitment strategies were used to ensure a broad, representative sample of HCPs with a range of experiences, current approaches, opinions and awareness regarding audio-vestibular conditions in patients with TBI. The survey invitation and link were circulated via professional organisations' contact lists including Headway UK, British Society of Physical and Rehabilitation Medicine, and Royal College of Speech and Language Therapists (see online supplemental appendix 2 for full list), promotion in blogs (e.g, Krysalis Neuro Occupational Therapy) and through social media channels (Facebook, Twitter, and Instagram). In addition, an invitation email with the survey link was sent to the HCPs identified via Expertscape website (https://expertscape.com/) as experts in the field of TBI. All of these distribution channels were selected taking into account the inclusion criteria of our study. Participants were not given a specific timeframe to complete the survey. Responses were obtained between May 2022 and December 2022. Participation in the survey was voluntarily and no incentives were provided.

## Data analysis

All survey data were imported to Microsoft Excel 2016. Prior to analysis, data were checked to ensure all respondents fully consented and responses were given to all questions as required. If any surveys were terminated early, they were not visible because they were not submitted. Only completed surveys were analysed. Descriptive statistics, in particular frequencies and percentages of responses to closed questions were calculated in SPSS (V.28). Open-text responses (short sentences or lists) were reviewed and categorised in Microsoft Excel 2016. In particular, similar responses were categorised together and then summarised.

Occupations, countries of residence, years of occupation, years of working with with TBI patients and hours per week working with TBI adults were analysed using nonparametric statistical methods to assess relations between these categorical variables. To ensure a meaningful statistical analysis in comparisons, countries of residence were divided into four groups (the UK, Europe, Asia and Others) and occupations into six groups as seen in table 1 (Group 1 representing physiotherapists; Group 2 occupational therapists; Group 3 speech and language therapists; Group 4 physiatrist or rehabilitation medicine doctors and rehabilitation nurses; Group 5 doctors from other specialties, and Group 6 'other') due to the imbalance in participant numbers. The Kruskal-Wallis test was used for comparing more than two independent groups in each category. Following this measurement, to identify the groups that contribute to the observed differences in the data, pairwise subgroup comparisons were conducted using Mann-Whitney U test with Bonferroni's correction for country of residence groups: $p < 0.05/(4 \times 3/2)$, for occupation groups: $p < 0.05/(6 \times 5/2)$, for years working with TBI adults groups: $p < 0.05/(4 \times 3/2)$.

## RESULTS
### Characteristics of participants

The characteristics of respondents are reported in table 1. A total of 70 HCPs completed the survey from 14 different countries. Of these, the majority were from the UK (n=30, 42.9%), followed by Türkiye (n=14, 20%) and New Zealand (n=8, 11.5%). Respondents reported a range of occupations, including physiotherapists (n=15, 21.4%), occupational therapists (n=11, 15.8%) and speech and language therapists (n=10, 14.2%). The majority (48/70, 69%) had been in their occupation for more than 10 years and over half (39/70, 56%) had been working with patients with TBI between 5 and 20 years. Over half of the respondents (57%) spent more than 10 hours per week working with adult patients with TBI. There were no statistically significant differences in years of occupation and years working with TBI adults among participants based on groups of country of residence and occupations (p>0.05). However, statistically significant differences were observed in the working hours per week with adults with TBI based on groups of country of residence and occupations in the Kruskal-Wallis test (p<0.05). Subgroup statistical analyses showed that participants from the UK (n=16, 23%) were more likely to spend more than 20 hours a week with TBI adults, but those from Asia (n=7, 10%) were more likely to spend less than 1 hour per week (p<0.008). Participants in Group 2 (occupational therapists; n=6, 9%) were reported to spend over 20 hours per week with adults with TBI, while participants in Group 5 (doctors from other specialties; n=5, 7%) spent less than 1 hour per week with adults with TBI (p<0.003). Exactly half of the respondents worked in private healthcare (35/70, 50%), and the other half worked in National Health Service or state-owned hospitals.

### The clinical experiences of HCPs regarding audio-vestibular consequences in patients with TBI

This section explores the reported and/or observed audio-vestibular symptoms in adult patients with TBI.

| Table 1 | Demographic information of participants | | |
|---|---|---|---|
| **Characteristics** | **n** | **%** | **Statistical analysis groups** |
| | 70 | 100 | |
| **Country of residence** | | | |
| UK | 30 | 42.9 | UK |
| Türkiye | 14 | 20.0 | Asia |
| New Zealand | 8 | 11.5 | Others |
| Ireland | 6 | 8.6 | Europe |
| Canada | 2 | 2.9 | Others |
| Australia | 2 | 2.9 | Others |
| Greece | 1 | 1.4 | Europe |
| USA | 1 | 1.4 | Others |
| Azerbaijan | 1 | 1.4 | Asia |
| Italy | 1 | 1.4 | Europe |
| Spain | 1 | 1.4 | Europe |
| Croatia | 1 | 1.4 | Europe |
| Venezuela | 1 | 1.4 | Others |
| Tanzania | 1 | 1.4 | Others |
| **Occupation** | | | |
| Physiotherapist | 15 | 21.4 | Group 1 |
| Occupational therapist | 11 | 15.8 | Group 2 |
| Speech and language therapist | 10 | 14.2 | Group 3 |
| Physiatrist or rehabilitation medicine doctor | 6 | 8.6 | Group 4 |
| Neuropsychiatrist | 5 | 7.1 | Group 5 |
| Neuropsychologist | 4 | 5.8 | Group 6 |
| Rehabilitation nurse | 4 | 5.8 | Group 4 |
| Neurologist | 3 | 4.2 | Group 5 |
| Psychologist | 2 | 2.9 | Group 6 |
| Geriatrician | 2 | 2.9 | Group 5 |
| Neurosurgeon | 2 | 2.9 | Group 5 |
| Optometrist | 1 | 1.4 | Group 6 |
| Case manager | 1 | 1.4 | Group 6 |
| Anaesthetist | 1 | 1.4 | Group 5 |
| Behavioural scientist | 1 | 1.4 | Group 6 |
| Primary care doctor | 1 | 1.4 | Group 5 |
| Orthopaedic surgeon | 1 | 1.4 | Group 5 |
| **Years in occupation** | | | |
| Less than 5 years | 8 | 11 | Less than 5 years |
| 5–10 years | 14 | 20 | 5–10 years |
| 10–20 years | 25 | 36 | 10–20 years |
| More than 20 years | 23 | 33 | More than 20 years |
| **Years working with TBI adults** | | | |
| Less than 5 years | 16 | 23 | Less than 5 years |

Continued

| Table 1 | Continued | | |
|---|---|---|---|
| **Characteristics** | **n** | **%** | **Statistical analysis groups** |
| 5–10 years | 20 | 29 | 5–10 years |
| 10–20 years | 19 | 27 | 10–20 years |
| More than 20 years | 15 | 21 | More than 20 years |
| **Hours per week working with TBI adults** | | | |
| Less than an hour | 7 | 10 | Less than an hour |
| 1–5 hours | 13 | 19 | 1–5 hours |
| 5–10 hours | 10 | 14 | 5–10 hours |
| 10–20 hours | 15 | 21 | 10–20 hours |
| More than 20 hours | 25 | 36 | More than 20 hours |

TBI, traumatic brain injury.

According to the results, the majority of responses ranged across 'a few' to 'most of my patients' describe the symptom (figure 1). In general, HCPs reported that 'some' to 'all' of their patients had auditory problems such as 'inability to understand speech-in-noise' (46/70, 66%), 'tinnitus' (45/70, 64%), 'hyperacusis' (40/70, 57%) and 'inability to capture conversations' (44/70, 63%) and balance problems such as 'dizziness' (55/70, 79%) and 'vertigo' (47/70, 67%). However, the majority of HCPs observed that 'none' to 'a few' patients reported problems with 'inability to walk in the dark without a physical problem' (31/70, 44%), 'inability to understand speech in quiet' (40/70, 57%) and 'hearing loss' (41/70, 58%). According to the Kruskal-Wallis test, no significant differences were found in the frequency of audio-vestibular symptoms reported by patients to HCPs, based on years of working with adults with TBI and groups of country of residence ($p>0.05$). Statistically significant differences were observed for symptom of 'inability to capture conversations' based on groups of occupations ($p<0.05$). According to the Mann-Whitney U test with Bonferroni correction, a statistically significant difference was detected between Group 3 (speech and language therapists) and Group 1 (physiotherapists) for the symptom of 'inability to capture conversations' ($p<0.003$). Most speech and language therapists (n=7, 10%) reported that 'most' of their patients described 'inability to capture conversations', while most physiotherapists (n=10, 14%) reported that only 'a few' of their patients reported this symptom.

The observed audio-vestibular symptoms by HCPs in patients with TBI (where trauma occurred more than 6 months ago) or reported to HCPs by patients' relatives are shown in figure 2. HCPs observed that 'some' to 'all' of their patients 'asked to repeat their speech frequently during their interviews' (45/70, 64%), 'difficulty understanding them on the phone' (38/70, 54%) and 'an inability to tolerate certain sounds' (44/70, 63%). In contrast, 37% of HCPs (26/70) did state that 'none' of their patients 'avoided coming to appointments because

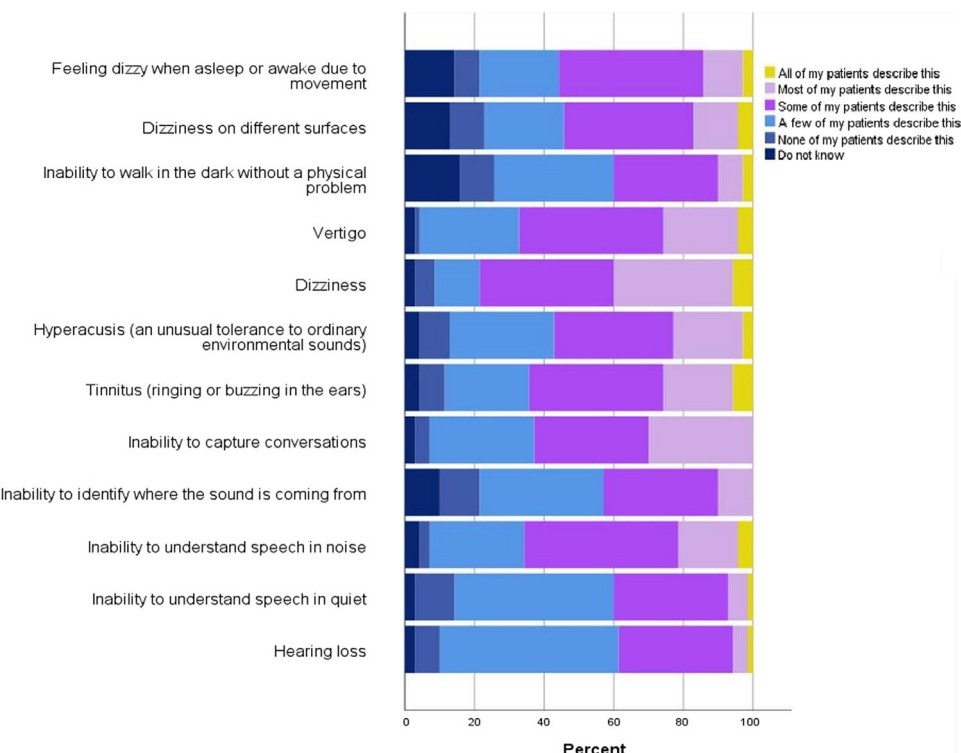

**Figure 1** The frequency of audio-vestibular symptoms reported by patients to healthcare professionals.

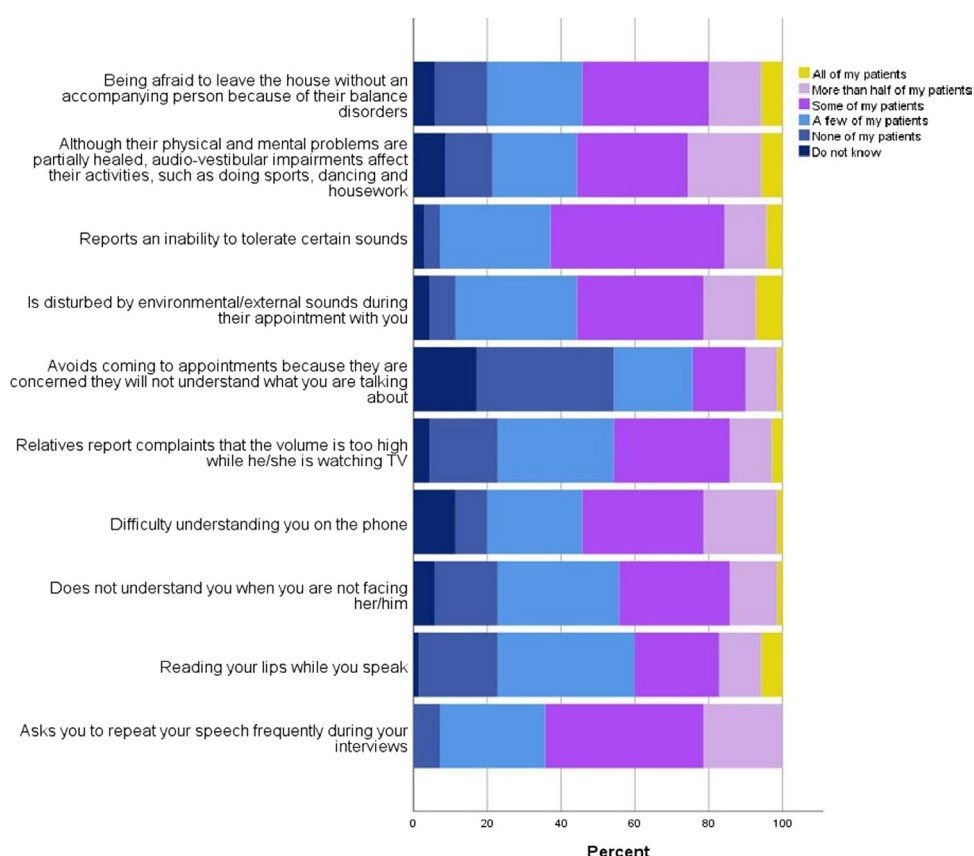

**Figure 2** The frequency of audio-vestibular symptoms reported by patients or their relatives to healthcare professionals and/or observed by healthcare professionals.

they were concerned, they would not understand what HCPs were talking about'. However, 17% (12/70) did not know whether there was such a concern for their patients. The details of the statistical analysis results regarding the associations between audio-vestibular symptoms observed by HCPs or reported by patients' relatives to HCPs, and the years working with adults with TBI and groups of country of residence, are explained in online supplemental appendix 3.

Statistically significant differences were observed in the Kruskal-Wallis test between groups of occupations regarding the symptoms of 'asking to repeat their speech frequently during their interviews', 'not understanding HCPs when they are not facing with them', 'difficulty understanding HCPs on the phone' (p<0.05). Subgroup statistical analyses demonstrated significant differences in communication-related symptoms between Group 1 (physiotherapists) and Group 3 (speech and language therapists) (p<0.003). Based on this, most speech and language therapists (n=7, 10%) reported that 'more than half' of their patients asked to repeat their speech frequently during interviews, while the majority of physiotherapists (n=12, 17%) indicated that 'some' to 'a few' of their patients asked to repeat their speech frequently. Additionally, most speech and language therapists (n=6, 9%) stated that 'some' of their patients did not understand them when they were not facing them, similar to physiotherapists where the same percentage (n=6, 9%) reported observing this problem in 'a few' of their patients. The majority of speech and language therapists (n=6, 9%) reported that 'some' of their patients had difficulty understanding them on the phone, while physiotherapists (n=4, 6% for each option) had an equal percentage distribution among the options 'some', 'a few' of their patients and 'do not know'. The statistical differences in other symptoms observed among other groups of occupations are detailed in online supplemental appendix 3.

Over 40% (30/70) of HCPs were unable to identify and generalise the aetiology of the patients with TBI who frequently reported audio-vestibular complaints. Of the 21% (15/70) that were able, most reported that likely aetiologies were vehicle-related collisions (9/15, 60%), followed by falls (2/15, 13%) and sports injuries (1/15, 7%). Those who selected 'other' trauma option (3/15, 20%) indicated that audio-vestibular complaints could result from any TBI aetiology, including an object falling on the head, vehicle-related collisions and/or sports-related injuries.

### The current approaches to assess and manage audio-vestibular conditions of patients with TBI

To understand the current approaches used by HCPs regarding the audio-vestibular conditions of their patients with TBI, we asked questions about screening and assessment methods. Over half of respondents (52/70, 74%) indicated that they ask their patients with TBI about their balance status (e.g. dizziness, vertigo), while the remaining stated that they only asked if their

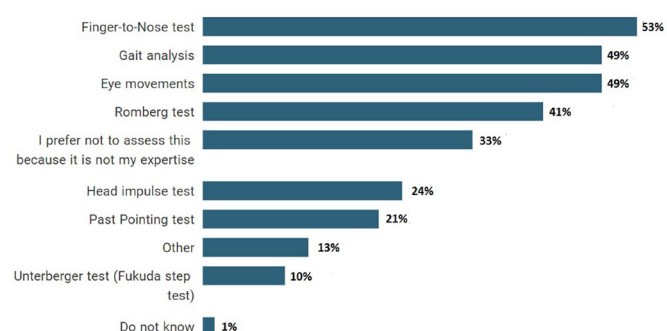

**Figure 3** Per cent of screening methods used by healthcare professionals when they notice dizziness or balance disorders in patients with traumatic brain injury.

balance status was interfering with the conduct of the appointment (10/70, 14%), they did not ask if there are no complaints (5/70, 7%) or they preferred not to ask because it was not their expertise (1/70, 1%). When they noticed dizziness or balance disorder in patients with TBI, the most commonly used screening methods were finger-to-nose (37/70, 53%), gait analysis (34/70, 49%) and eye movements (34/70, 49%) as seen in figure 3. In addition to these tests, HCPs also reported using evaluation methods such as videonystagmography (VNG) and dynamic visual acuity in open-text responses. Again, several HCPs reported that they prefer not to use any screening method because it was not their area of expertise (23/70, 33%). In relation to checking the external ear condition of patients with TBI, 13% (9/70) of HCPs, in particular physiatrists or rehabilitation medicine doctors reported that they evaluated 'the external ear' by themselves with an otoscope. However, almost half of the respondents (34/70, 49%) stated that they would refer patients to the ENT and/or audiology department to check the external ear conditions.

When asked how many patients with TBI they overall referred to ENT and/or audiology, 78% (55/70) of HCPs said that they referred 'some' or 'a few' of their patients with TBI. Only 4% (3/70) of HCPs referred 'all' of their patients with TBI to ENT and/or audiology department. In the questions asked to understand the opinions and awareness of HCPs, HCPs stated that they considered referring patients with TBI to the ENT specialist and/or audiology department, when their patients could not understand and/or hear their speech (37/70, 53%), when their patients reported tinnitus and hyperacusis (33/70, 47%) or when they observed vertigo and/or dizziness in their patients (29/70, 41%). Only 6% (4/70) felt that audio-vestibular conditions could be ignored because patients with TBI struggled with many impairments (figure 4). In contrast, when asked whether patients with TBI should use a hearing aid if they have hearing loss, no one thought it was something that could be ignored. However, most of the respondents (31/70, 44%) stated that patients with TBI with hearing loss should use hearing aids 'if they would like to', rather than 'definitely', while the remaining did not know (14/70, 20%). Among the

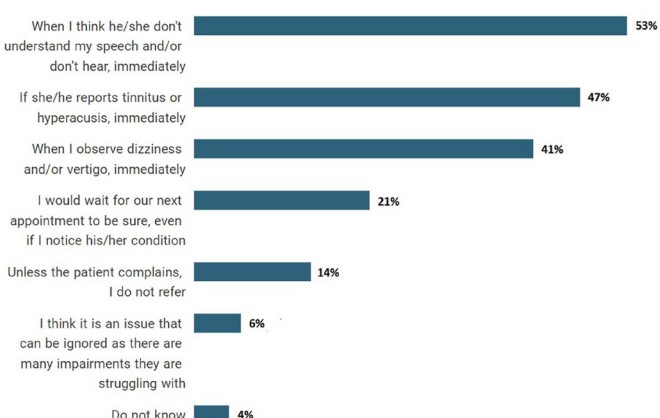

**Figure 4** Per cent of reasons for healthcare professionals to refer patients with traumatic brain injury to the ENT and/or audiology department.

participants who indicated not knowing the majority were physiotherapists (n=8, 11%). When asked if they could help their patients with TBI using hearing aids at their appointments, interestingly while 40% (28/70) stated that they did not know anything about hearing aids, 40% (28/70) also stated that they could change the battery of the hearing aid. The majority (32/70, 46%), however, did state that they would notice whether the hearing aid was working. Notably, speech and language therapists were not among the HCPs who reported not knowing anything about hearing aids.

## DISCUSSION

This study explored the clinical experiences, current approaches, opinions and awareness of HCPs of audio-vestibular consequences in patients with TBI. In general, the majority of HCPs had observed audio-vestibular impairments in their patients with TBI in line with the literature.[17–19] However, most HCPs reported that hearing loss was only observed in 'a few' of patients. In contrast to this finding, a systematic review study which investigated peripheral auditory dysfunction secondary to TBI showed that sensorineural hearing loss was the most frequently observed auditory impairment at a rate of 37.3% in patients with non-blast related TBI.[17] Additionally, Knoll et al[20] showed that following TBI, individuals can experience auditory symptoms even if their hearing thresholds are within the normal range (≤25 dB HL). Therefore, these studies suggest that hearing loss can be observed or reported more frequently by patients with TBI than HCPs may observe. One potential explanation for these differences could be that the HCPs who participated in this study are not experts in audio-vestibular assessments and as such may not be looking for audiological impairments, particularly during appointments which may be addressing other concerns. However, recognition of hearing loss in patients with TBI by non-audiological-specialist HCPs may result in more onwards referrals

for detailed assessment and input, resulting in better outcomes.

We also observed that HCPs were more likely to assess vestibular and/or balance-related conditions of their patients with TBI than their hearing-related conditions. One explanation for this is that different HCPs such as neurologists, physiotherapists and occupational therapists, who participated in this study, may be more likely to assess and treat balance and/or vestibular function, and less likely to be familiar with auditory assessments. Another explanation could be that these professional groups prioritise physical function. These factors might be a possible reason why patients with auditory problems were more likely to be referred. Additionally, the observation of communication-related symptoms more likely in individuals with TBI by speech and language therapists rather than physiotherapists can be associated with their field of expertise in communication. These findings may also contribute to emphasising the importance of differences in patient care among professional groups and a multidisciplinary approach.

However, in contrast, a number of HCPs did not refer to ENT or audiology services if the patient had not complained about auditory problems and/or if they felt that the audio-vestibular problems were an issue that were not a priority compared with other TBI-related impairments. This suggests that some HCPs may not be aware of the negative short-term and long-term consequences of untreated audio-vestibular impairments. Many HCPs reported that they would leave the use of a hearing aid to the discretion of individual patients. This is understandable, in patients who do not want to use a hearing aid are unlikely to use one, even if it is provided. However, it may also reflect lack of training and education in hearing aids for HCPs. It is important that HCPs explore the patient's rationale and capacity[21] for making this decision as well as explaining the potential consequences of untreated hearing loss.[12]

Finally, consistent with the studies in the literature,[15 22] several HCPs reported that patients with TBI with audio-vestibular complaints had the most vehicle-related collision aetiology. However, this was not universal and likely reflects the fact that audio-vestibular complaints are a common consequence of a wide variety of TBI aetiologies.

Interestingly, a recent Australian study[15] also showed that audiologists lacked knowledge about the assessment and management of audio-vestibular impairments of patients with TBI in audiological services. While there can be variation in understanding of audio-vestibular consequences of TBI across clinical services and/or centres around the world, this study demonstrated a similar lack of awareness regarding the audio-vestibular consequences of patients with TBI in non-audiology HCPs. This highlights the evident need for greater awareness among HCPs working with this patient population, even if it is outside their area of expertise. For this purpose, it may be beneficial to prepare a practical audiology guide for all HCPs dealing with TBI in future studies.

## Strengths and limitations

To our knowledge, this is the first study to investigate the clinical experiences, current approaches, opinions and awareness of HCPs, other than ENT specialists and audiologists, regarding the audio-vestibular consequences of patients with TBI. This international survey received responses from a total of 70 participants from 17 different departments and 14 countries and as such we have gained knowledge from a range of HCPs who are actively working in clinical settings that assess and manage audio-vestibular consequences in patients with TBI. However, we do recognise that the majority were from the UK and that there are low to no participation rates for some countries. This means that some of the findings may reflect the UK health system above others. For example, other countries may have audiologists/ENTs as part of their multidisciplinary team. Open online surveys may have inherent issues such as the possibility of participants submitting multiple responses and leading to response fraud in recruitment through particularly social media. However, anonymity with this method allows respondents to provide honest and candid answers freely.[23] Although no tracking method was used to address these issues in our study, the shared social media accounts and links included individuals from the academic community and HCPs. In addition, responses were checked for inconsistencies or evidence of deception through data cleaning. Furthermore, the inclusion of HCPs such as physiotherapists working in the field of balance may have influenced the results, in particular the confidence in vestibular assessments. Finally, we wanted to understand the experience of HCPs who specialise in TBI, therefore we did not include audiologists and/or ENT specialists in this survey and as such we do not know their experiences with this patient population, except for the findings from the Australia survey[15] which indicate similar experiences to our findings here. Future research should consider further investigation into the experiences of audiologists/ENT specialists in relation to this patient population, in particular the assessments that should be carried out.

## Conclusion

HCPs who care for individuals who sustained TBI observe a wide range of audio-vestibular outcomes such as inability to understand speech-in-noise, tinnitus, hyperacusis, dizziness and vertigo in these patients. HCPs' current approaches, opinions and awareness around further assessment, investigation and treatment were variable in this study. HCPs may be more likely to assess vestibular and/or balance-related conditions than their hearing-related conditions in this patient group. In general, HCPs were referring only some or a few of their patients with TBI to ENT and/or audiology department which may negatively impact the potential diagnosis of audio-vestibular (especially auditory) impairments in patients with TBI. Untreated audio-vestibular impairments may have significant adverse consequences for patients and non-expert clinicians may be unaware of these. We suggest that standardised screening protocols may enable easier identification of audio-vestibular problems with signposting to appropriate specialties for further assessment and intervention. Thus, audio-vestibular impairments of patients with TBI can be diagnosed and treated earlier, to improve quality-of-life of patients with TBI .

**Author affiliations**
[1]Hearing Sciences, Division of Mental Health and Clinical Neuroscience, School of Medicine, University of Nottingham, Nottingham, UK
[2]National Institute of Health and Social Research (NIHR) Nottingham Biomedical Research Centre, Nottingham, UK
[3]Division of Rehabilitation Medicine, University Hospitals of Derby and Burton NHS Foundation Trust, Derby, UK
[4]Injury, Inflammation and Recovery Sciences, School of Medicine, University of Nottingham, Nottingham, UK
[5]School of Healthcare Enterprise and Innovation, University of Southampton, Southampton, UK

**Acknowledgements** The authors wish to acknowledge the clinicians who piloted the survey and all the respondents of the survey. We would like to express our gratitude to Professor David Baguley, who sadly passed away during this study. His knowledge and expertise helped develop the fundamental ideas for this research, for which we are very thankful.

**Contributors** DMB, LE, KB authors conceptualised the project and design of the survey questions. KB led the recruitment and data collection, supported by KF, DMB, LE. KF, DMB and LE informed methodology and interpretation of analysis. KB conducted initial analysis. KF and LE reviewed and refined analysis. KB drafted manuscript. KF, LE and KB reviewed and approved the final manuscript for submission. KF is responsible for the overall content as the guarantor.

**Funding** This work was supported by the Ministry of National Education of the Republic of Türkiye and the National Institute for Health Research (NIHR) Nottingham Biomedical Research Centre (N/A for the award/grant number).

**Disclaimer** KB was funded by the Ministry of National Education of the Republic of Türkiye to undertake this work as part of her PhD (N/A for the award/grant number). KF was funded by National Institute for Health Research (NIHR Post-Doctoral Fellowship, PDF-2018-11-ST2-003) at the time of completing this work. The views expressed in this publication are those of the author(s) and not necessarily those of the NIHR, the NHS or the Department of Health and Social Care.

**Competing interests** None declared.

**Patient and public involvement** Patients and/or the public were not involved in the design, or conduct, or reporting or dissemination plans of this research.

**Patient consent for publication** Not applicable.

**Ethics approval** Research approval was obtained from the University of Nottingham, Faculty of Medicine and Health Sciences Research Ethics Committee (FMHS 462-0222, May 2022). All respondents gave informed consent at the beginning of the survey. Participants gave informed consent to participate in the study before taking part.

**Provenance and peer review** Not commissioned; externally peer reviewed.

**Data availability statement** Data are available upon reasonable request. Data are available upon reasonable request. Summarised data relevant to the study are included in the article or uploaded as supplemental information (online supplemental data). Raw data can also be shared by the authors upon reasonable request.

**ORCID iDs**
Kübra Bölükbaş http://orcid.org/0000-0002-7625-4758
Laura Edwards http://orcid.org/0000-0003-1392-2648
David M Baguley http://orcid.org/0000-0002-0767-0723
Kathryn Fackrell http://orcid.org/0000-0001-6529-8643

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
