## [Reviewer comments · BMJ Open]

ARTICLE DETAILS

TITLE (PROVISIONAL)	The Clinical Experiences, Current Approaches, Opinions and Awareness of Healthcare Professionals regarding the Audio-Vestibular Consequences of Individuals with Traumatic Brain Injury: A Cross-sectional Online Survey Study
AUTHORS	Bölükbaş, Kübra; Edwards, Laura; Baguley, David; Fackrell, Kathryn

VERSION 1 – REVIEW

REVIEWER	Nada, Ebtessam Hamed Zagazig University
REVIEW RETURNED	11-Oct-2023

GENERAL COMMENTS	First topic is very interesting Line 284 taking about poor audiologist experience in dealing with audio-vestibular complaints in TBI patients Author can give a hint about the audio vestibular medicine practice like that gives in Edinbrugh and Egypt where audio-vestibular physicians are well aware of any audio-vestibular sequel in most disorders. https://www.rcpe.ac.uk/
--

REVIEWER	Liu, Bo Huazhong University of Science and Technology, Otorhinolaryngology-Head Neck Surgery
REVIEW RETURNED	15-Oct-2023

GENERAL COMMENTS	This manuscript investigated the current status of clinical experiences, current approaches, opinions, and awareness of the healthcare providers (HCPs) who care for adults with traumatic brain injury (TBI) regarding the audio-vestibular consequences (other than ENT specialists and audiologists) by using Internet survey. The authors found that the majority of TBI patients encountered had auditory and vestibular complaints, which may be overlooked by non-expert HCPs. A simple and standardized screening protocol is needed to improve identification and management of audio-vestibular problems in these TBI patients. Some major and minor issues still need to be improved: 1. The response rate should be given2. The statistical method should be detailed in the Data Analysis section.3. The percentage of occupational duration when dealing with TBI patients for 5-20 years needs to be recalculated. According to Table 1, this percentage equals to 56% (39/70).4. Does the respondent characteristics influence the clinical practice, such as duration of practice, geographic area?
--

REVIEWER	Franklin, Chu University of Buea, Zoology and Animal Physiology
REVIEW RETURNED	20-Dec-2023

GENERAL COMMENTS	Comments to Authors The article addresses audio-vestibular consequences in individuals who sustained traumatic brain injury (TBI): They outline the awareness of a diverse range of health care professionals (HCPs) excluding ENT/Audiology experts, on audio-vestibular problems in TBI patients (6 months at least) post-injury, their experiences on the different audio-vestibular consequences observed and also their awareness in the assessment of these problems. To achieve these aims, the authors conducted an online cross-sectional survey involving 14 countries across the continents. They received 70 responses which is considered a small sample for this type of study. However, the study is a multi-country study and have responses from health care professionals from 14 countries which may have greatly improved the inputs of the work. The authors make important findings on the types of audio-vestibular conditions observed, they also report how these TBI patients are assessed for these problems and if they referred or not to ENT/Audiology departments. The authors emphasize on the need for none ENT/Audiology experts to be trained on how to assess and detect these audio-vestibular problems as they have adverse consequences on TBI patients if not detected and treated accordingly. To this effect they recommend a protocol be established to be used by none ENT/Audiology HCPs in the course of managing TBI cases. I find the work novel and intriguing as most often, the audio-vestibular problems in TBI seem to be overlooked on the profit of physical and cognitive problems. I therefore congratulate the authors for this piece of work which would add knowledge to the scientific community. However, the work needs to be improved on several aspects noted below: Abstract I think there should be a brief introduction (background) before the objectives Page 3, line 29: "Objective". Lines 32, 33: I suggest study design and participants be explained under the "methods" section of the abstract The study design and participants should be presented under the methods section. I think the authors should include a phrase on how data was analysis (software used). Line 41, the authors write "the majority from the UK" (include the proportion that matches this declaration). Line 44: "In general, HCPs.... Also, the authors did not report results on the awareness of the HCPs on the audio-vestibular consequences after TBI, but their conclusion was focused on this aspect. Page 4, line 59: I suggest the authors should take the strengths and weaknesses of the study to "after the discussion just before the conclusions" Introduction Generally, the introduction is well written and organized, however, find a few minor comments below: Lines 75, 76: I suggest the definition of TBI be rephrased, the word trauma is coming three times...the authors could use the definition
--

of the national institute of health (NIH, 2018); "Traumatic brain injury (TBI) is an alteration in brain function, or other evidence of brain pathology, caused by an external force
I think the citations could be separated by commas; "e.g
"...behavioural, and audio-vestibular impairments.5,6,7,8

Methods

Line 147: I think it is important for the authors to mention the targeted HCPs at this level

Line 151: I suggest the authors mention the number of HCPs who finally participated in the survey here, was there a predefined targeted sample? Did the authors reach their targeted sample? If not, are there reasons why? Also, the inclusion and exclusion criteria is not clearly brought out, were HCPs with no clinical experience included? Were all questionnaires submitted considered? I think it is imperative to state clearly the detailed inclusion and exclusion criteria.

Line 157: Considering it was an online survey, were dispositions taken to prevent double answering (one HCP submitting more than one filled questionnaire) (response fraud), especially for those that responded from social media network like Facebook, twitter or who were not directly invited through email? I suggest the authors include a statement on this, as this aspect is one of the inconveniences of online surveys

Results

Lines 176 to 180: I suggest the authors include the percentages next to the numbers in the bracket, "UK (30, 43%), Turkey (14, 20%)" for example

Line 182: the authors write "over 50% spent more than ...". Their exact proportion has not been mentioned. I suggest a rephrase to "Over half (proportion or %) of the respondents spent more than 10 hours...."

Line 184: Full meaning of NHS in bracket

Line 185: I suggest an a row be added in Table between "Characteristics and Country of residence" like below

Characteristics n %

n 70 100

Country of residence

Line 198: Insert Figure 1

Line 201: I suggest "HCPs observed that some to all of their "patients" "asked" to repeat their speech....

Line 203: the authors write "inability to tolerate certain sounds"...what could be the characteristics of such sound? High pitch?, I think...the sentence could be completed by adding "certain sounds like...."

Line 213: insert figure 2

Line 223: I suggest rephrase as "...and eye movements (34/70, 49%) as seen in Figure 3.

Line 223: Please include the proportion of HCPs that used videonystagmography (VNG) and dynamic visual acuity

Line 228: suggest to write "...whilst 13% (9/70) reported that they evaluated 'the external ear by themselves' with...."

230: Insert Figure 3

Result presented in Line 227 about the proportion of HCPs referring to ENT/Audiology departments seems contradictory to that reported in line 231...I think the authors should look into it, may be line 227 needs to be more precised

Line 244: the words "knew" or "did not know" should be taken off...

Line 247: insert figure 4

	The quality of the figures could be improved for clarity. Also, the axis of the figures are not completely defined for Figures 1 and 2 and completely absent for figure 3 and 4 Discussion 249; it better reads “This study...” 251: “In our findings”....should replace “In our study” 257 to 258, needs a rephrase, not understandable 259 to 261: The authors states the possible explanations to the differences in the results with the systematic review (20), and they write “the HCPs who participated in this study are not experts in audio-vestibular assessment” but I was wondering, in the systematic review, those who observed these audio-vestibular problems, were there experts? If so, then it is important to state this point. Lines 264 to 269: I think this paragraph could be improved, by explaining why these HCP seem to overlook the hearing difficulties, ...probably because they prioritize physical function, and feel it is the most important deficit... As a general comment in this section, I suggest authors do not use more of personalising words ...like “Our study” which is coming so many times,...”This study, our findings, the results obtained from this study indicates....etc. could be better replacements I suggest Response bias should be added as limitation, that is if measures were not taken to avoid this. I also think another limitation could be that the sample is small, may be in further studies, more strategies could be employed to have more responses, the study was not precised in Europe and America, as I find one African Country in the list, which means the questionnaires might not have reached most countries Conclusion 310: I suggest “HCPs who care for individuals who sustained TBI observe....” 312 to 314: the authors write “Untreated audio-vestibular impairments may have significant adverse consequences for patients and non-expert clinicians may be unaware of these” ...I think the authors should conclude regarding the awareness before this statement could follow...Also, I think it is important for authors to state a conclusion on the most frequent audio-vestibular problems observed in individuals who sustained TBI. It should be seen that the all objectives (aspects) investigated have a conclusion
--	---

VERSION 1 – AUTHOR RESPONSE

Reviewer 1. Dr. Ebtessam Hamed Nada, Zagazig University, Zagazig University Faculty of Human Medicine

Point 1. Line 284 taking about poor audiologist experience in dealing with audio-vestibular complaints in TBI patients -Author can give a hint about the audio vestibular medicine practice like that gives in Edinburgh and Egypt where audio-vestibular physicians are well aware of any audio-vestibular sequel in most disorders.

Response 1. Thank you for your comments. We have revised the discussion to include a sentence on the variation in understanding the audio-vestibular consequences of TBI.

'Whilst there can be variation in understanding of audio-vestibular consequences of TBI across some clinical services and/or centres around the world, this study demonstrated a similar lack of awareness regarding the audio-vestibular consequences of TBI patients in non-audiology HCPs.' (Page 14, Lines 363-366).

Reviewer 2. Dr. Bo Liu, Huazhong University of Science and Technology Point

1. The response rate should be given

Response 1. Thank you for your valuable comments. The study was conducted online, and we used a range of recruitment strategies, including emails to professional networks. This meant we were unable to record response rates as individual invitations were not sent to all potential participants.

Point 2. The statistical method should be detailed in the Data Analysis section.

Response 2. Thank you for your comment. We have stated that we calculated frequencies and percentages of responses to closed-text questions. We have also added more detail on the open-text data analysis and additional statistical methodology as seen in Response 4.

Point 3. The percentage of occupational duration when dealing with TBI patients for 5-20 years needs to be recalculated. According to Table 1, this percentage equals to 56% (39/70).

Response 3. We have corrected this.

Point 4. Does the respondent characteristics influence the clinical practice, such as duration of practice, geographic area?

Response 4. Thank you for your question. A range of statistical methods and detailed evaluations were made to determine whether there are differences between occupational groups, country of residence, working hours, years of working with TBI, and working hours per week with TBI adults. The statistical measurements made are explained in detail in both the data analysis section and the results of the article.

For example, ‘There were no statistically significant differences in years of occupation and years working with TBI adults among participants based on groups of country of residence and occupations ($p>0.05$). However, statistically significant differences were observed in the working hours per week with TBI adults based on groups of country of residence and occupations in the Kruskal-Wallis test ($p<0.05$). Subgroup statistical analyses showed that participants from the UK ($n=16$, 23%) were more likely to spend more than 20 hours a week with TBI adults, but those from Asia ($n=7$, 10%) were more likely to spend less than 1 hour per week ($p<0.008$). Participants in Group 2 (occupational therapists; $n=6$, 9%) were reported to spend over 20 hours per week with TBI adults, whilst participants in Group 5 (doctors from other specialties; $n=5$, 7%) spent less than 1 hour per week with TBI adults ($p<0.003$) (Page 8, Lines 209-219).’

Lines for all other additions for statistical measurements: (Page 7, Lines 186-199; Page 10, Lines 234-244; Page 11, Lines 252-272)

However, we think it is important to emphasise that this study was not statistically powered to answer questions regarding this sort of subgroup analysis and so any findings must be interpreted with caution. For the same reason, while we have reported some of these statistical subgroup analyses in the main text, we have left several in the supplemental information (Supplemental Appendix 3).

Reviewer: 3. Dr. Chu Franklin, University of Buea

Abstract.

Point 1. I think there should be a brief introduction (background) before the objective. -Page 3, line 29: “Objective”. - Lines 32, 33: I suggest study design and participants be explained under the methods” section of the abstract. The study design and participants should be presented under the methods section.

Response 1. Thank you for your valuable comments. The abstract followed the format required by BMJ Open and therefore cannot be changed.

Point 2. I think the authors should include a phrase on how data was analysis (software used).

Response 2. The software used for data analysis has been added to the abstract section as suggested.

Point 3. Line 41, the authors write “the majority from the UK” (include the proportion that matches this declaration).

Response 3. We have added the proportion to this declaration.

Point 4. Line 44: "In general, HCPs...."

Response 4. We have corrected this typo.

Point 5. Also, the authors did not report results on the awareness of the HCPs on the audio-vestibular consequences after TBI, but their conclusion was focused on this aspect.

Response 5. We have revised the results section in the abstract to highlight the results regarding the awareness of HCPs on audio-vestibular consequences after TBI to align with our insights reported in the conclusion.

'For auditory impairments, HCPs preferred referring TBI patients to Audiology/ENT services. However, 6% of HCPs felt that audio-vestibular conditions could be ignored upon referral because TBI patients struggled with many impairments. Additionally, 44% would suggest hearing aids to TBI patients with hearing loss 'if they would like to use' rather than 'definitely.' (Page 2, Lines 45-49)

Point 6. Page 4, line 59: I suggest the authors should take the strengths and weaknesses of the study to "after the discussion just before the conclusions"

Response 6. This section is in accordance with BMJ Open requirements.

Introduction.

Point 7. Lines 75, 76: I suggest the definition of TBI be rephrased, the word trauma is coming three times...the authors could use the definition of the national institute of health (NIH, 2018); "Traumatic brain injury (TBI) is an alteration in brain function, or other evidence of brain pathology, caused by an external force.

Response 7. We have rephrased the sentence as suggested.

'Traumatic brain injury (TBI) is defined as a structural injury and/or physiological deterioration of brain function following trauma due to an external force' (Page 4, Lines 75-76).

Point 8. I think the citations could be separated by commas; "e.g "....behavioural, and audio-vestibular impairments.5,6,7,8

Response 8. Thank you for your suggestion. Commas were added when the reference format was changed to a Vancouver-style square bracket, as recommended by BMJ Open.

Methods.

Point 9. Line 147: I think it is important for the authors to mention the targeted HCPs at this level.

Response 9. We have added examples of targeted HCPs to the 'Patient and Public Involvement' section.

'The study focused on the experiences of HCPs in the field of TBI, such as physiotherapists, neuropsychologists, rehabilitation nurses,...'(Page 6, Lines 154-155).

Point 10. Line 151: I suggest the authors mention the number of HCPs who finally participated in the survey here.

Response 10. Thank you for the suggestion. However, this section is describing the survey methodology, and we feel that the number of respondents to the survey should be reported in the results section as we did know how many HCPs would participate.

Point 11. Was there a predefined targeted sample? Did the authors reach their targeted sample? If not, are there reasons why?

Response 11. We did not set a specific target for participant numbers in our cross-sectional study. As this was an exploratory survey and the first conducted internationally in this field with other healthcare professionals, we did not have enough evidence to determine approximately how many participants would be needed.

Point 12. Also, the inclusion and exclusion criteria is not clearly brought out, were HCPs with no clinical experience included? I think it is imperative to state clearly the detailed inclusion and exclusion criteria.

Response 12. We have revised the study sample section and emphasised the inclusion and exclusion criteria.

'The study population was identified as HCPs caring for adult patients with TBI worldwide. To be eligible to participate in the survey, respondents were required to be HCPs with clinical experience in caring for adult patients with TBI and to have the ability to understand and answer questions in English (except HCPs in Türkiye). ENT

specialists and audiologists were excluded from the study as they were already experts in the field of audio-vestibular care.’ (Page 6, Lines 160-164).

Point 13. Were all questionnaires submitted considered?

Response 13. All questionnaires submitted were considered. Before analysis, all responses were reviewed to ensure that respondents had fully consented and that all questions had been answered.

Point 14. Line 157: Considering it was an online survey, were dispositions taken to prevent double answering (one HCP submitting more than one filled questionnaire) (response fraud), especially for those that responded from social media network like Facebook, twitter or who were not directly invited through email? I suggest the authors include a statement on this, as this aspect is one of the inconveniences of online surveys.

Response 14. Thank you for the suggestion. As recognised double answering is one of the inconveniences of online surveys. We tried to account for this by primarily recruiting HCPs through collaboration with associated professional organisations and networks, and personalised e-mails to HCPs identified via the Expertscape website. However, we understand the inherent challenges in ensuring complete exclusivity, especially in the context of social media recruitment. The social media accounts shared and the links to these accounts included people from the academic community and HCPs. In addition, through data screening, we checked for inconsistencies in individual responses or evidence of deceit. We have highlighted these issues with online surveys in the limitations section in the discussion.

‘Open online surveys may have inherent issues such as the possibility of participants submitting multiple responses and leading to response fraud in recruitment through particularly social media. However, anonymity with this method allows respondents to provide honest and candid answers freely.²³ Although no tracking method was used to address these issues in our study, the shared social media accounts and links included individuals from the academic community and HCPs. In addition, responses were checked for inconsistencies or evidence of deception through data cleaning’ (Page 15, Lines 380-386)

Results.

Point 15. Lines 176 to 180: I suggest the authors include the percentages next to the numbers in the bracket, “UK (30, 43%), Turkey (14, 20%)” for example.

Response 15. We have added the percentages.

Point 16. Line 182: the authors write “over 50% spent more than ...”. Their exact proportion has not been mentioned. I suggest a rephrase to “Over half (proportion or %) of the respondents spent more than 10 hours....”

Response 16. We have revised this sentence.

'Over half of the respondents (57%) spent more than 10 hours per week working with adult patients with TBI.' (Page 8, Lines 208-209).

Point 17. Line 184: Full meaning of NHS in bracket.

Response 17. We added meaning of NHS.

Point 18. Line 185: I suggest an a row be added in Table between "Characteristics and Country of residence" like below

Characteristics	n	%
N	70	100
Country of residence		

Response 18. We have added a row to the table as suggested.

Point 19. Line 198: Insert Figure 1

Response 19. Following BMJ Open requirements, Figure 1 will be inserted as point of publication.

Point 20. Line 201: I suggest "HCPs observed that some to all of their "patients" "asked" to repeat their speech...."

Response 20. We have edited grammatical errors.

Point 21. Line 203: the authors write "inability to tolerate certain sounds"...what could be the characteristics of such sound? High pitch?, I think...the sentence could be completed by adding "certain sounds like...."

Response 21. The survey response option used "inability to tolerate certain sounds" and did not include an explanation as it could be a number of different sounds that could cause problems (certain sounds may vary from person to person). As we want

to maintain the same text as provided to HCPs we have not changed this within the text.

Point 22. Line 213: insert figure 2

Response 22. Following BMJ Open requirements, Figure 2 will be inserted as point of publication.

Point 23. Line 223: I suggest rephrase as "...and eye movements (34/70, 49%) as seen in Figure 3.

Response 23. We have rephrased this.

Point 24. Line 223: Please include the proportion of HCPs that used videonystagmography (VNG) and dynamic visual acuity

Response 24. Thank you for your recommendation. Since this was based on open text responses, we could not specify the percentages. However, we have clarified in the text that these tests were provided in open responses.

'In addition to these tests, HCPs also reported using evaluation methods such as videonystagmography (VNG) and dynamic visual acuity in open-text responses.' (Page 12, Lines 289-291).

Point 25. Line 228: suggest to write "...whilst 13% (9/70) reported that they evaluated 'the external ear by themselves' with...."

Response 25. It has been rewritten.

Point 26. Line 230: Insert Figure 3

Response 26. Following BMJ Open requirements, Figure 3 will be inserted as point of publication.

Point 27. Result presented in Line 227 about the proportion of HCPs referring to ENT/Audiology departments seems contradictory to that reported in line 231...I think the authors should look into it, may be line 227 needs to be more precised.

Response 27. These are two different questions. The first sentence reports the result regarding the question 'How do you check the external ear condition of your adult

patients with traumatic brain injury (which occurred more than 6 months ago)?'. The following sentence reports the rates of individuals with TBI they refer based on the question 'How many of your adult patients who have had a traumatic brain injury (which occurred more than 6 months ago) do you refer to the audiology or ENT department?' We rewrote these parts to ensure clarity.

'In relation to checking the external ear condition of TBI patients, 13% (9/70) of HCPs, in particular physiatrists or rehabilitation medicine doctors reported that they evaluated 'the external ear' by themselves with an otoscope. However, almost half of the respondents (34/70, 49%) stated that they would refer patients to the ENT or audiology department to check the external ear conditions.' (Page 12, Lines 292-296)

'When asked how many TBI patients they overall referred to ENT and/or audiology, 78% (55/70) of HCPs said that they referred 'some' or 'a few' of their TBI patients. Only 4% (3/77) of HCPs referred 'all' of their TBI patients to ENT and/or audiology department.' (Page 12, Lines 298-300)

Point 28. Line 244: the words "knew" or "did not know" should be taken off...

Response 28. We have removed 'knew.'

Point 29. Line 247: insert figure 4.

Response 29. Following BMJ Open requirements, Figure 4 will be inserted as point of publication.

Point 30. The quality of the figures could be improved for clarity. Also, the axis of the figures are not completely defined for Figures 1 and 2 and completely absent for figure 3 and 4

Response 30. Thank you for your suggestion. The quality of the figures was prepared according to BMJ Open standards. We added necessary explanations and revised below each figure.

'Figure 3. Percent of screening methods used by HCPs when they notice dizziness or balance disorders in patients with TBI.

'Figure 4. Percent of reasons for HCPs to refer TBI patients to the ENT and/or audiology department.' (Page 21, Lines 536-539)

Discussion.

Point 31. Line 249; it better reads “This study...”

Response 31. It has been changed.

Point 32. Line 251: “In our findings”...should replace “In our study”

Response 32. This sentence was revised.

‘However, most HCPs reported that hearing loss was only observed in ‘a few’ of patients’ (Page 13, Lines 322)

Point 33. Line 257 to 258, needs a rephrase, not understandable.

Response 33. It has been rephrased.

‘Therefore, these studies suggest that hearing loss can be observed or reported more frequently by TBI patients than HCPs may observe.’ (Page 13, Lines 327-329)

Point 34. Line 259 to 261: The authors states the possible explanations to the differences in the results with the systematic review (20), and they write “the HCPs who participated in this study are not experts in audio-vestibular assessment” but I was wondering, in the systematic review, those who observed these audio-vestibular problems, were there experts? If so, then it is important to state this point.

Response 34. Thank you for your notification. The review did not specifically provide the expertise of those who observed problems. We have clarified in the text the purpose of the review.

Point 35. Lines 264 to 269: I think this paragraph could be improved, by explaining why these HCP seem to overlook the hearing difficulties, ...probably because they prioritize physical function, and feel it is the most important deficit...

Response 35. Thank you for your suggestion. We have highlighted as well that HCPs prioritize physical function. However, considering that this survey did not explicitly inquire about whether balance problems are perceived as the most significant deficit by HCPs, to prevent potential misleading of readers we did not wish to state this as fact.

‘We also observed that HCPs were more likely to assess their TBI patients’ vestibular and/or balance-related conditions than their hearing-related conditions. One explanation

for this is that different HCPs such as neurologists, physiotherapists and occupational therapists, who participated in this study, may be more likely to assess and treat balance and/or vestibular function, and less likely to be familiar with auditory assessments. Another explanation could be that these professional groups prioritize physical function. These factors might be a possible reason why patients with auditory problems were more likely to be referred.’ (Page 13, Lines 335-341)

Point 36. As a general comment in this section, I suggest authors do not use more of personalising words ...like “Our study” which is coming so many times,...”This study, our findings, the results obtained from this study indicates....etc. could be better replacements

Response 36. Thank you for this attention. They have been changed.

Point 37. I suggest Response bias should be added as limitation, that is if measures were not taken to avoid this. I also think another limitation could be that the sample is small, may be in further studies, more strategies could be employed to have more responses, the study was not precised in Europe and America, as I find one African Country in the list, which means the questionnaires might not have reached most countries

Response 37. Thank you for your suggestion. We added an explanation about this limitation in strength and limitation’s part.

‘However, we do recognise that the majority were from the UK and that there are low to no participation rates for some countries. This means that some of the findings may reflect the UK health system above others. For example, other countries may have audiologists/ENTs as part of their multi-disciplinary team.’ (Page 15, Lines 377-380).

Conclusion.

Point 38. Line 310: I suggest “HCPs who care for individuals who sustained TBI observe....”

Response 38. We have revised.

Point 39. Line 312 to 314: the authors write “Untreated audio-vestibular impairments may have significant adverse consequences for patients and non-expert clinicians may be unaware of these” ...I think the authors should conclude regarding the awareness before this statement could follow...

Response 39. Thank you for your comment. An explanation was added in the results section to make the awareness results clearer:

'In the questions asked to understand the opinions and awareness of HCPs, HCPs stated that they considered referring TBI patients to the ENT specialist and/or audiology department...'. (Page 12, Lines 300-302)

In the conclusion section, a revision was made as well to reveal the conclusion regarding awareness.

'HCPs' opinions and awareness around further assessment, investigation and treatment were variable in this study'. (Page 15, Lines 397-398).

Point 40. Also, I think it is important for authors to state a conclusion on the most frequent audio-vestibular problems observed in individuals who sustained TBI.

It should be seen that the all objectives (aspects) investigated have a conclusion

Response 40. Many thanks for your suggestion. Considering the diverse nature of audio-vestibular impairments observed in individuals with TBI, there was not a single frequently observed problem. Therefore, in the conclusion section, we needed to emphasize the wide range of audio-vestibular problems encountered by individuals who have sustained TBI.

VERSION 2 – REVIEW

REVIEWER	Liu, Bo Huazhong University of Science and Technology, Otorhinolaryngology-Head Neck Surgery
REVIEW RETURNED	05-Mar-2024

GENERAL COMMENTS	The revised version reads well and I suggest acceptance.
--

REVIEWER	Franklin, Chu University of Buea, Zoology and Animal Physiology
REVIEW RETURNED	08-Mar-2024

GENERAL COMMENTS	Comments to Authors The article addresses audio-vestibular consequences in individuals who sustained traumatic brain injury (TBI): They outline the awareness of a diverse range of health care professionals (HCPs) excluding ENT/Audiology experts, on audio-vestibular problems in TBI patients (6 months at least) post injury, their experiences on the different audio-vestibular consequences observed and also their awareness in the assessment of these problems. To achieve these aims, the authors conducted an online cross-sectional survey involving 14 countries across the continents. Most of the comments in the previous review were addressed. The paper reads better. However, see below some minor comments
---

	#1. The conclusion should reflect the findings. The objective of the study was to “To explore the experiences, current approaches, opinions, and awareness of healthcare professionals (HCPs) caring for adults with Traumatic Brain Injury (TBI) regarding the audio-vestibular consequences”. It should be seen that the conclusion of the paper, is addressed with respect to the objective. #2. I suggest, since the strengths and weaknesses are stated already just after the abstract, below, before the conclusion, authors should write just on the “limitations”
--	--

VERSION 2 – AUTHOR RESPONSE

Reviewer: 3. Dr. Chu Franklin, University of Buea

Point 1. The conclusion should reflect the findings. The objective of the study was to “To explore the experiences, current approaches, opinions, and awareness of healthcare professionals (HCPs) caring for adults with Traumatic Brain Injury (TBI) regarding the audio-vestibular consequences”. It should be seen that the conclusion of the paper, is addressed with respect to the objective.

Response 1. Thank you for your comments. We have revised the conclusion as suggested.

Point 2. I suggest, since the strengths and weaknesses are stated already just after the abstract, below, before the conclusion, authors should write just on the “limitations”.

Response 2. Thank you for your thoughtful suggestion. The “Strengths and Weaknesses” section following the abstract aligns with the requirements of BMJ Open. The strengths and limitations in the discussion section includes novelty and other strengths of the study that are not included in the summary after the abstract, therefore we have to keep the current title.